# Inclusion of Natural Antioxidants of Mango Leaves in Porous Ceramic Matrices by Supercritical CO_2_ Impregnation

**DOI:** 10.3390/ma15175934

**Published:** 2022-08-27

**Authors:** María del Cisne Guamán-Balcázar, Antonio Montes, Diego Valor, Yorky Coronel, Desireé M. De los Santos, Clara Pereyra, Enrique J. Martínez de la Ossa

**Affiliations:** 1Department of Chemical Engineering and Food Technology, Faculty of Sciences, University of Cádiz, International Excellence Agrifood Campus (CeiA3), 11510 Puerto Real, Spain; 2Departamento de Química, Universidad Técnica Particular de Loja, San Cayetano Alto sn, AP, Loja 1101608, Ecuador; 3Department of Physical Chemistry, Faculty of Sciences, University of Cádiz, International Excellence Agrifood Campus (CeiA3), 11510 Puerto Real, Spain

**Keywords:** supercritical impregnation, natural antioxidants, porous silica, mango leaves

## Abstract

Mango is one of the most important, medicinal tropical plants in the world from an economic point of view due to the presence of effective bioactive substances as co-products in its leaves. The aim of this work was to enhance the impregnation of natural antioxidants from mango leaves into a porous ceramic matrix. The effects of pressure, temperature, impregnation time, concentration of the extract and different porous silica on impregnation of phenolic compounds and antioxidant activity were analyzed. The volume of the pressurized fluid extract and amount of porous ceramic matrix remained constant. The best impregnation conditions were obtained at 6 h, 300 bar, 60 mg/mL, 35 °C and with MSU-H porous silica. The results indicated that increasing the pressure, concentration of the extract and temperature during impregnation with phenolic compounds such as gallic acid and iriflophenone 3-C (2-O-p-hydroxybenzolyl)-β-D-glucoside increased the antioxidant activity and the amount of total phenols.

## 1. Introduction

Mango (*Mangiferina indica* L.) from the Anacardiaceae family is a native fruit from Southeast Asia which is grown mainly in tropical regions in more than 87 countries [1]. Its exploitation produces a high amount of agro-industrial waste, such as seeds, peels, leaves and steam bark, derived from the activity of pruning and industrial fruit processing. In traditional medicine, several uses have been reported for the different parts of the mango plant; the skin has been used in the treatment of diphtheria and rheumatism, the cooked fruit in the treatment of diarrhea and chronic dysentery, the fresh leaves to tone the gums, the smoke of the leaves in the treatment of ailments of the throat and the ash for scalds and burns [2].

There have been many investigations that showed the antioxidant, antimicrobial, antidiabetic and anticarcinogenic activity of mango and its by-products [3,4]. Martínez et al. [5] and Meneses et al. [6] determined a high number of phenolic compounds and high amount of antioxidant capacity in mango skin extracts and precipitates, identifying compounds such as mangiferin, isomangiferin, quercetin, kaempferol and quercetin glycosides. On the other hand, Fernández-Ponce et al. [7] determined its cytotoxic effect in breast cancer cell lines, attributing their results to the synergistic effects of the different polyphenols, such as mangiferin, gallotannins, methyl gallate and homomangiferin, present in the extract, while Abdullah et al. [8] studied the anticancer effect of mango seed extract and of mango kernel on estrogen-receptor-positive human breast carcinoma cells (MCF-7), determining that the extract induced apoptosis of cancer cells through the activation of oxidative stress.

Currently, the industry in general is searching for green processes to obtain active principles; therefore, the use of supercritical CO_2_ is an alternative for the extraction, precipitation, encapsulation and impregnation of compounds with biological activity. Regarding the impregnation process, impregnation using supercritical fluids is a sustainable alternative to incorporating active substances in polymeric and porous ceramic matrices due to the fact that it offers several advantages compared to other conventional impregnation techniques, such as immersion, the processes of which require a greater amount of solvent and elimination by evaporation processes and can affect temperature sensitive compounds. Likewise, when the porous ceramic matrix comes into contact with the organic solvent, a compact mass is created, losing the physical characteristics and manageability of the porous ceramic matrix. Moreover, unwanted reactions between substances, heterogeneous dispersion and low percentages of impregnation are the other drawbacks of conventional impregnation [9].

The supercritical solvent Impregnation (SSI) process is based on the use of porous and polymeric matrices, as well as gels, to incorporate bioactive compounds in a sustainable way through the use of a solvent such as CO_2_, which solubilizes the substance at supercritical pressure and temperature, and, due to its high diffusivity, it penetrates the matrix, achieving the impregnation of the phenolic compounds [10]. Specifically, in the impregnation process in porous ceramic matrices, the solvent (CO_2_) and the solute at supercritical conditions of pressure and temperature diffuse on the surface of the pores of the matrix during the depressurization process. The amount of impregnated substance depends, on the one hand, on the process conditions, which are tuned and, thus, vary the substance loading and depth of impregnation [9]. On the other hand, the amount of impregnated substance depends on the specific surface area of the matrix, so the higher area, the greater the impregnation of the active principle. Moreover, solvent-free final products are achieved due to CO_2_ being released as a gas after depressurization [9].

Most phenolic compounds have polar characteristics, and, for this reason, co-solvents such as ethanol are used in order to increase the polarity of the system (CO_2_ + antioxidants) and, thus, increase the percentage of impregnation [11,12].

There are several pieces of research where the supercritical solvent impregnation process was used to impregnate bioactive compounds. Cejudo et al. [13] impregnated antioxidant compounds from European Olea leaf ethanolic extract in polyethylene terephthalate/polypropylene (PET/PP) films and concluded that, at high pressures (300 and 400 bar), the plastic impregnated with the ethanolic extract showed a higher antioxidant activity than the plastic impregnated with pure caffeic acid. On the other hand, Sanchez-Sanchez et al. [9] used CO_2_/methanol (50%) extract to impregnate antioxidant compounds in polyester textile.

Several researchers used porous silica for drug impregnation using supercritical solvent impregnation. García-Casas et al. [14] used pure quercetin to impregnate mesoporous silica beads SB-300; a ZnO mesoporous nanostructure was impregnated with ibuprofen, clotrimazole and hydrocortisone [15]; piroxicam was used on mesoporous silica such as SBA-15 and Grace Syloid^®^XDPd [16]; cinnamon essential oil was used on Zein/MCM-41 nanocomposite film [17]; rhodium acetylacetonate was used on mesoporous silica such as MCM-41, MSU-H and HMS [18]; vitamin E acetate was used on silica MCM-41 [19]; and palladium nanoparticles were used on SBA-15 [20]. For this reason, and on the basis of the research presented above, the goal of this investigation was to enhance the impregnation of natural antioxidants from mango leaves extract into a mesoporous silica by the supercritical solvent impregnation process. These impregnated silicas can be used as a drug-delivery formulation in the cosmetic and pharmaceutical fields or as implants in biomedicine. The effect of pressure, temperature, mango leaves extract concentration, impregnation time and type of silica on the number of antioxidants that were incorporated into the porous ceramic matrix was analyzed.

## 2. Materials and Methods

### 2.1. Materials

Mesoporous silica beads SB-300 were purchased from Miyoshi Europe laboratories as impregnation support. Spherical silica beads SB-300 have a specific surface area of 300 m^2^/g and average particle size of about 5 µm [14,21]. Likewise, in order to compare the amount of impregnation, a second MSU-H mesostructured silica was used with a pore size of 7.1 nm, 750 m^2^/g specific surface area, 0.91 cm^3^/g pore volume and 11.6 nm mean particle size, purchased from Sigma Aldrich (Hamburg, Germany). 

2,2-Dyphenil-1-picrylhydrazyl (DPPH), Folin-Ciocalteu phenol reagent (2N), gallic acid, ±-6-Hydroxy-2,5,7,8-tetramethylchromane-2-carboxylic acid (Trolox) (≥97%), 2, 4, 6-Tris (2-pyridyl)-striazine (≥98%), ferric chloride hexahydrate (≥98%), penta-O-galloyl-β-D-glucose hydrate (≥96%), quercetin and mangiferin (≥98%), 3,4-dihydroxybenzoic acid (≥97%) and quercetin 3-D-galactoside were purchased from Sigma-Aldrich (Steinheim, Germany). Sodium carbonate, sodium acetate trihydrate, hydrochloric acid (37%) and glacial acetic acid were purchased from Merck (Darmstadt, Germany). Acetonitrile and formic acid (HPLC grade), ethanol (98%), sodium chloride, monobasic potassium phosphate and sodium hydroxide were supplied by Panreac (Barcelona, Spain). CO_2_ (99.8%) was supplied by Linde (Barcelona, Spain). Milli-Q grade water with double distillation was used.

*Mangifera indica* L. leaves (*Kent* variety) were collected in 2019 by Finca Experimental ‘La Mayora’ Superior Centre of Scientific Research (CSIC) (Málaga, Spain) and dried until a drying loss of 91% was achieved. The leaves were crushed until they reached an average particle diameter of 750 µm.

### 2.2. Obtaining of Ethanolic Extracts from Mango Leaves

The mango leaves extract (20 mg/mL) was prepared by pressurized fluid extraction (PLE) of 60 g of mango leaves over three hours in a SF100 pilot plant developed by Thar Technologies (Pittsburgh, PA, USA). In the extraction process, 250 mL of ethanol as a co-solvent was used at 120 bar of pressure, 80 °C and 10 g ethanol/min flow rate [22].

### 2.3. Supercritical Solvent Impregnation (SSI)

An SSI pilot plant (Thar Technologies), the scheme of which is shown in Figure 1, was used for the impregnation of antioxidant compounds of mango leaves into silica. The plant included a CO_2_ bottle, a chiller to cool the CO_2_ and keep the fluid in a liquid state prior to entering the high-pressure pump for CO_2_ and a heat exchanger to heat the CO_2_ before introducing it into an impregnation cell of high pressure. In this cell, 5 mL of ethanolic extract and a porous metal basket of 10 µm, in which 200 mg of silica was contained, were introduced. Batch impregnation process was performed for the time established in each experimentation, and there was no direct contact between the extract and the porous silica.

The effects of five independent variables, pressure (100–350 bar), temperature (35–50 °C), time of impregnation (6–22 h), ethanolic extract concentration (20–80 mg/mL) and type of silica, on the impregnation process were evaluated while amount of silica (200 mg), volume of extract (5 Ml), stirring rate (400 rpm) and depressurization rate (90 bar/min) were kept constant throughout the process. These levels of variables were selected according to the experience [12] and limitation of the equipment and the degradability of the extract. A summary of the experiments carried out is detailed in Table 1. Moreover, the step order and sequence of these experiments are indicated in Figure 2.

### 2.4. Determination of Antioxidant Capacity Assay with DPPH and FRAP

#### 2.4.1. DPPH Method

The ability of the antioxidant compounds (donors of hydrogen or an electron) from mango leaves extract, present in the porous silica, to reduce the free radicals of the reagent DPPH was determined according to the method described by Bastante et al. [13], Scherer and Godoy [23] and Brand-Williams et al. [24]. In the samples with a low percentage of impregnation (experiments at different times and pressures), the percentage of DPPH that reacted with the silica impregnated was measured for 40 mg of impregnated material previously dried in an oven at 40 °C for approximately 30 min. A 4 mL amount of the DPPH solution (6 × 10^−5^ mol/L ethanol) was used, and the absorbance reading was performed at 515 nm. The impregnation percentage was obtained using the following Equation (1) [13]:(1)$$\%I=\frac{Abs_{0}-Abs_{t}}{Abs_{0}}*100$$
where *Abs*_0_ is the absorbance of the DPPH reagent, and *Abs_t_* is the absorbance after 3 h of reaction between the impregnated material and the DPPH. 

The antioxidant capacity, through the procedure established by Scherer and Godoy [23], was applied to the material with the highest level of impregnation (experiments at different concentrations, temperatures and types of silica). A 5 mL amount of ethanol was added to 101 mg of impregnated material, and it was placed in the ultrasound equipment for 30 min; then, the extract was filtered. To determine the antioxidant capacity, 0.1 mL aliquots of the extract were taken at six different concentrations, and 3.9 mL of 6 × 10^−5^ mol/L ethanol solution was added; it was left to react for 3 h, and the absorbance reading was carried out at 515 nm. Concentrations between 250 and 1000 ppm were used for the extract and between 20 and 2000 ppm for the impregnated material.

To calculate the antioxidant capacity index (*AAI*) (Equation (2)), it was necessary to determine the *IC*_50_, which was obtained by means of the extract concentration vs. DPPH remaining. It was calculated considering the absorbance of *DPPH* at 3 h of reaction (*CDPPH_t_*) and the initial concentration of *DPPH* (*CDPPH*_0_) according to the following equation:(2)$$AAI=C_{DPPHt\;}IC_{50}$$
(3)$$\%DPPH\;remaining=\frac{CDPPH_{t}}{CDPPH_{0}}\;\times \;100$$

A standard curve for *DPPH* was prepared to quantify the antioxidant capacity. Different concentrations between 7 × 10^−5^ and 6 × 10^−6^ mol/L ethanol were used. The results were expressed in μg/mL. 

On the other hand, the ability of antioxidants to trap the radical *DPPH* (2,2-diphenyl-1-picryhydrazyll) was also determined using the Trolox reagent as standard [24,25]; readings were made in duplicate in a 1.5 mL glass cell, and absorbance at 515 nm was measured on a 7305 UV/Visible spectrophotometer, JENWAY^®^. The results were expressed as equivalent micromoles of Trolox per gram of extract (µmol TE/g of extract).

#### 2.4.2. FRAP Method

This method was applied based on the method of Benzie et al. [26] but with various modifications from Thainpong et al. [25], which allowed us to quantify the iron reduction power in the extracts using Trolox (6-hydroxy-2,5,7,8-tetramethylchroman-2-carboxylic acid) as a standard. A 150 µL amount of extract (200 ppm) and 2850 µL of working solution (300 mM acetate buffer, 10 mM TPTZ solution in 40 mM HCl and 20 mM FeCl_3_·6H_2_O solution) were mixed for 30 min. Readings were taken in duplicate in a 1.5 mL glass cell, and absorbance at 593 nm was measured on a 7305 UV/Visible spectrophotometer, JENWAY ^®^. The results were expressed as equivalent micromoles of Trolox per gram of extract on a wet basis (µmol TE/g of extract).

### 2.5. Phenolic Compounds Determination in the Impregnated Silica

#### 2.5.1. Total Phenolic Content

The Folin–Ciocalteu colorimetric method based on that detailed by Swain and Hillis [27] with modifications from Thaipong et al. [25] was used to determine the content of total phenols in the different samples using gallic acid as a standard. A 150 µL amount of extract at a concentration of 200 ppm was mixed with 2400 µL of distilled water and 300 µL of 1N Na_2_CO_3_ and allowed to react for 2 h. Readings were made in duplicate in a 1.5 mL glass cell, and absorbance at 725 nm was measured on a JENWAY ^®^ 7305 UV/Visible spectrophotometer calibrated with distilled water. Results were expressed as gallic acid equivalent milligrams (mg GAE/g).

#### 2.5.2. HPLC Analysis

The determination and quantification of the impregnated phenolic compounds were performed by liquid chromatography (HPLC) (Agilent HPLC system 1100 series (Agilent, Germany)). HPLC was coupled to a UV/Vis detector and controlled by ChemStation^®^ HP software. The column used for separation was a Hydro-RP C18 reversed phase column (Phenomenex, Torrance, CA, USA) with a 4.0 mm × 2.0 mm i.d. C18 ODS guard column. The process was run at 25 °C using 0.1% (*v*/*v*) formic acid in water (phase A) and 0.1 % (*v*/*v*) formic acid in acetonitrile as mobile phase (phase B). The gradient program was as follows (phase B): 0 min, 0%; 0.2 min, 0%; 0.3 min, 7%; 14.7 min, 8.5%; 40 min, 19%; 45 min, 33%; 148 min, 50%; 50 min, 95%; 57 min, 0%; 63 min, 0%. The volume of each extract that was used was 20 µL, and mobile phase flow was 0.6 mL/min, and the compounds were detected at 278 nm. All analyses were carried out in triplicate.

### 2.6. Physical Characterization of Impregnated Silica

The sizes and morphologies of the samples were evaluated using a Nova NanoSEMTM 450 scanning electron microscope (SEM) that incorporated energy-dispersive X-ray spectroscopy (EDX). For SEM, the samples were deposited on carbon tape and covered by a 15 nm gold film. To determine particle size, SEM images were processed using image software (Scion Image from Scion Corporation, Chicago, IL, USA). EDX was carried out to analyze the chemical composition of different spheres. The samples were then deposited directly on aluminum support without carbon tape to avoid any carbon interferences. On the other hand, to determine any change in the surface of the impregnated silica particles, transmission electron microscopy (TEM) in JEOL2100 LaB6 was performed. A Bruker Tensor 37 FTIR spectrophotometer with a spectral resolution of 0.6 cm^−1^ was used to carried out the Fourier-transform infrared (FTIR) spectroscopy with the aim to identify possible chemical bond variation during the impregnation process. The spectra were resolved in a 4000–400 cm^−1^ range. KBr technique was used to measure the transmittance with potassium bromide pellets containing 1% weight of sample.

### 2.7. Statistic Analysis

Antioxidant capacity (AAI) and number of phenolic compounds data were analyzed using Microsoft Excel. For the discrimination of mean (*p* ≤ 0.05), a one-way analysis of variance (ANOVA) was performed using the LSD test. The analyses of different pressures (100–300 bar), concentrations (20–80 mg/mL), temperatures (35–50 °C) and types of silica (MSU and SB-300) were carried out in duplicate, and the standard deviation (SD) was calculated in each case.

## 3. Results and Discussion

### 3.1. Chemical and Functional Characterization of Extract 

The extract was obtained by pressurized fluids using ethanol as solvent, and the yield obtained was 18.75%. Regarding the antioxidant activity index (AAI) of the extract, the value was higher than 2, which, according to the classification carried out by Scherer and Godoy [23], can be considered a strong antioxidant capacity. On the other hand, the antioxidant capacity measured by the DPPH and FRAP method was 2637.46 and 1898.78 µmol TE/g extract, respectively, as shown in Table 2. 

When comparing the results with other investigations, the antioxidant capacity measured by the DPPH and FRAP method was higher than that of Tanacetum vulgare (4.69 and 4.55 µmol TE/g sample) and Juglans regia (1.19 and 1.28 µmol TE/g sample) [28] and lower than that of Moringa oleifera (602.2 and 229.6 µmol TE/g sample) [29] and Manguifera indica leaves (494.52 and 356.02 µmol TE/g sample) [30].

Regarding the content of total phenols, the amount that was obtained in the present investigation, 279.53 ± 3.27 mg GAE/g extract, was higher than that of leaves extracts obtained from Moringa oleifera [31], citrus × aurantium [32], Azadirachta indica [33], Stevia rebaudiana [34], Olea europaea [35], Tanacetum vulgare [28] and Manguifera indica [30,36], as shown in Figure 3. 

Eleven compounds were identified, of which ten were quantified, highlighting that the majority of compounds were those such as iriflophenone 3-C-β-D-glucoside, mangiferin, gallic acid and iriflophenone 3-C-(2-O-*p* -hydroxybenzoyl)-β-D-glucoside, as can be seen in Table 3 and Figure 4. These compounds had between five and ten times the concentration of the rest of compounds and were responsible for the antioxidant activity. These compounds were also determined by Fernández et al. [4] using a high-pressure technique to produce enriched, potent, antioxidant phenolic compound extracts and by Guamán-Balcázar et al. [37] in those extracts obtained by conventional extraction. 

### 3.2. Impregnation Process—Time Influence

Initially, six impregnation times (3, 6, 9, 12, 15, 18 and 22 h) were evaluated, keeping pressure (100 bar), temperature (35 °C), extract concentration (20 mg/mL), stirring rate (400 rpm) and depressurization rate (100 bar/min) constant, as indicated in Figure 5. 

From the obtained results, it can be observed that, when increasing the impregnation time from 3 to 6 h, there was an increase of approximately 11% in the impregnation percentage; however, after 22 h of the process, there was a decrease of approximately 15% in the impregnation percentage. García-Casas et al. [14] spent 2 h impregnating quercetin into mesoporous silica beads SB-300, while Bouledjouidja et al. [16] determined that 4 h was the effective length of time for impregnating compounds, such as vitamin E, into OMS-7 porous silica. 

In contrast, some authors, such as García-Casas et al., also reported that increasing the impregnation time increases the impregnation percentage [12]. Thus, authors determined that, at 24 h, the antioxidant load of mango leaves in porous silica SB-300 was higher. This fact suggests that the impregnation time depends, on the one hand, on the nature of the solute and its solubility in CO_2_, and, sometimes, it only takes a short time for the antioxidant compounds to diffuse over the porous ceramic matrix. On other hand, interactions that are produced between molecules of the surface of the pores and the fluid should be taken into account. Pore size distributions, surface area and pore volume are the main parameters required in order to predict any results. Most of the porous networks are formed by interconnected pores with irregular shapes, like a pore labyrinth, which lead to a different tortuosity, and even these textural data are not able to reflect it [38]. 

Regarding the chemical composition of the impregnated silica at different times, the phenolic compounds could not be quantified; however, six compounds were identified as gallic acid, methyl gallate, iriflophenone 3-C-β-D-glucoside, iriflophenone 3-C-(2-op-hydroxybenzole)-β-D -glucoside, mangiferin and iriflophenone 3-C-(2-o-galloryl)- β-D -glucoside. In Figure 6, it can be seen that, at 6 h, gallic acid and iriflophenone 3-C-(2-o-p-hydroxybenzol)-β-D-glucoside were mainly found, while compounds such as methyl gallate and mangiferin were not identified.

### 3.3. Impregnation Process—Pressure Influence

The effect of operating pressure on impregnation percentage was investigated using different pressures from 100 to 350 bar, at 35 °C, 20 mg/mL ethanolic extract concentration, 200 mg of silica SB-300, 100 bar/min depressurization rate and 400 rpm stirring rate. It can be observed in Table 4 that the number of compounds, such as gallic acid and iriflophenone 3-c-(2-op-hydroxybenzole)-β-d-glucoside, increased when pressure was increased. This fact can be explained by the solvation and diffusivity capacity of CO_2_ + ethanol, which increase with pressure; thus, the solubility of phenolic compounds in the supercritical solution increases and, thus, increases the impregnation percentage. 

Therefore, the observed effects cannot be explained in terms of solubility limitations, but, instead, can be explained in terms of CO_2_-drug-silica interactions, mainly CO_2_-drug and drug-silica interactions. The first is represented by the solubility at a given temperature and pressure and can be described in terms of solvent density and drug volatility. The second is more specific and depends on the drug chemical structure and the presence of functional groups capable of interacting with the hydroxyl groups of the silica surface (silanol groups). These factors determine the partition coefficient of the drug between the CO_2_ (solvent phase) and the silica particles’ surface which, ultimately, determines the total impregnation/deposition yield.

When the experiment was performed at higher pressure, the CO_2_ density and its solvent power were higher than at low pressure (at constant temperature). Therefore, when equilibrium is reached, CO_2_-drug interactions are favored, and a lower amount of the drug is expected to be adsorbed on the particles’ surface.

On the other hand, the depressurization step decreases solvent density and drug solubility, leading to drug precipitation or deposition. In this way, the drug is partly impregnated or adsorbed onto the particles’ surface and partly deposited, with no specific interaction with the surface. When depressurization starts at a higher pressure level and/or when it is performed at a slow rate, more of the drug is removed (re-dissolved in the solvent phase) before it begins to precipitate on the silica particles. This explains why the best impregnation/deposition yields were generally obtained at conditions with a lower pressure (120 bar) and a faster depressurization rate (10 bar/min) and the lowest yields with the opposite conditions (250 bar and 5 bar/min).

### 3.4. Impregnation Process—Concentration of the Extract and Temperature Influence

To evaluate the influence of the extract concentration on the impregnation process (Table 3), experiments were carried out with concentrations (Ce) of 20 to 80 mg/mL and with constant impregnation time (6 h), pressure (300 bar), temperature (35 °C) and type of silica (SB-300). In Table 5, it can be observed that 20 mg/mL and 80 mg/mL were the lowest values obtained for antioxidant capacity index (AAI), while, when increasing the Ce from 40 to 60 mg/mL, the AAI increased (*p* < 0.05). Regarding the percentage of impregnation between 20 and 60 mg/mL (90.23 ± 2.64–87.70 ± 0.31), there was no significant difference (*p* > 0.05); however, when working with 80 mg/mL, the values decreased significantly. Thus, it can be presumed that, at concentrations greater than 60 mg/mL, the silica became saturated, and, for this reason, the AAI values were less than 80 mg/mL. The AAI values obtained from the impregnated material were approximately ¼ of the antioxidant capacity of the pressurized liquid extract (PLE1). According to the classification defined by Scherer and Godoy [23], the impregnated silica had a moderate antioxidant capacity. Sanchez-Sanchez et al. [9], in the impregnation of mango leaf extract into a polyester fabric, determined an antioxidant capacity index of AAI: 1.3 under operating conditions of 400 bar/35 °C and of AAI: 4 under operating conditions of 500 bar/55 °C. These values obtained were higher than those of the present investigation. With respect to the impregnated phenolic compounds in Table 5, it can be observed that the concentration of the extract affected the impregnation of phenolic compounds, that is, when the concentration of the extract increased from 20 to 80 mg/mL (*p* < 0.05), there was an increase in compounds such as gallic acid and iriflophenone 3-C-(2-o-p-hidroxibenzol)-β-D-glucoside.

Regarding the temperature variable, once the time (6 h), pressure (300 bar), depressurization rate (100 bar/min) and extract concentration (20 mg/mL) had been defined and fixed, two temperatures (35 °C and 50 °C) and two different silicas (SB-100 y MSU-H) were assayed to determine the influence of these variables in the impregnation process. At low temperatures and at high pressures, the density of CO_2_ increased and, with it, the ability to solubilize the active substances; on the contrary, as the temperature increased, the scCO_2_ compounds’ solubility increased by reducing the vapor pressure. In our case, the first effect prevailed; thus, the increase in temperature from 35 °C to 50 °C had a negative effect on the antioxidant capacity of the impregnated matrix, that is, under isobaric conditions of 300 bar (Table 5) and at a temperature of 50 °C, the density of CO_2_ decreased, and the solubility of phenolic compounds was consequently reduced. Murga et al. [39] determined a decrease in solubility from 1.2 × 10^−7^ to 1.9 × 10^−8^ mole fraction when increasing the temperature from 40 °C to 60 °C.

### 3.5. Impregnation Process—Type of Silica

On the other hand, the effect of the different silicas was studied; when using MSU–H, the impregnation of phenolic compounds was greater (*p* < 0.05) than in silica SB-300. Likewise, it can be observed that the number of total phenols in silica MSU-H was greater than in SB-300. A similar trend was determined in the antioxidant capacity measured by the FRAP method; however, using DPPH, a greater amount was obtained when working with silica SB-300. In Table 5, it can be observed that MSU-H was the silica into which the highest quantity of antioxidants was impregnated at 35 °C (AAI; 1.05 ± 0.13). Authors such as Ushiki et al. [18] and Dao et al. [40] determined that the impregnation of compounds depends on the pore size and the specific surface of the material being impregnated; therefore, it seems that the greater impregnation of antioxidant compounds in MSU-H was caused by the specific surface area (750 m^2^/g), which was higher than that of silica SB-300 (300 m^2^/g).

The IR patterns of the MSU and SB-300 silicas before and after mango leaves impregnation were assessed in order to identify the main functional groups contained in the samples and their possible interactions (Figure 7). The wide, broad band in the region between 3700 and 3200 cm^−1^ belonged to the stretching of the surface silanol groups of O–H bonds, together with stretching bands of hydrogen-bonded water molecules. The strong peak around 1090–1010 cm^−1^ was the asymmetrical stretching vibrations band of siloxane, –Si–O–Si–. The symmetrical stretching vibration of –Si–O–Si– was around 800 cm^−1^, and its bending mode appeared around 480 cm^−1^. A low band at 990–945 cm^−1^ was referred to the Si–O bond stretching of the silanol group. The deformational vibrations of physiosorbed water molecules were reflected by a peak around 1650–1600 cm^−1^ [41]. Iriflophenone, gallic acid and mangiferin were the main identified compounds in the impregnated silica. Thus, the carbonyl groups present in these compounds interacted with the silanol groups or with the molecules of water adsorbed on the silica. In any case, the carbonyl stretching band did not appear in the spectra due to the low levels of impregnation. A peak at 1398 cm^−1^, corresponding to the stretching of C–H, was the only indication of the deposition of polyphenols onto both silicas.

### 3.6. Morphology of Particles

The impregnated particles were analyzed by SEM and TEM. In Figure 8, it can be seen that there was no difference in the images with respect to the shape of the particles between the non-impregnated (a) and impregnated (b) SB-300 silica. Both the silica particles and the precipitated particles produced by the SAS process of mango leaf antioxidants were spherical particles [22,37], which made it difficult to notice the difference between images (a) and (b) and to show the impregnation area with them; however, the size of the D50 particle of the impregnated silica (0.40 µm) was smaller than that of the silica (0.85 µm). Thus, it is possible that the reduction in D50 was due to the fact that there were smaller particles of antioxidants from the mango leaves adsorbed into the particles of the silica.

Additionally, in the particle size distribution graphs it can be observed that the particles were homogeneous; however, in the impregnated silica, there was a higher frequency of particles from 0.1 to 0.5 µm. In this regard, energy-dispersive X-ray spectroscopy was carried out to obtain the relationship between composition and the size of spheres (Figure 9). Carbon present in polyphenols was the only element that could be discerned between the polyphenols and silica. Analyses were carried out directly on the aluminum support without carbon film to avoid carbon interferences. In both samples, a prominent aluminum peak from support was precisely shown. However, contrary to expectations, it was observed that smaller spheres did not present carbon in their composition, but carbon was found when higher spheres were analyzed. Thus, impregnated compounds were deposited on the pore surface of the silica spheres. In this sense, the carbon peak in the spectrum was not so high due to the silica but exhibited a higher proportion than impregnated compounds. Moreover, the TEM images (Figure 10) did not indicate a clear difference between silica alone and impregnated silica, but it seems that antioxidant compounds were adsorbed on the matrix due to the pore size and specific surface area of the matrix. The difference between the particles was not so evident, but the texture of the impregnated sample appeared to be coarser.

## 4. Conclusions

In this investigation, the impregnation of natural antioxidants into porous ceramic matrices, such as SB-300 and MSU-H silicas, was successfully achieved. Supercritical impregnations were carried out using different variables, such as pressure, temperature and impregnation time, as well as extract concentration, in order to increase the impregnation percentage. Among all the variables, concentration and silica were the variables that most influenced the antioxidant capacity; likewise, the main impregnated compound was iriflophenone 3-C-(2-o-p-hydroxybenzole)-β-D-glucoside.

The impregnated silica had a moderate antioxidant capacity; thus, this material could be used in the cosmetic industry as a source of natural antioxidants and in biomedicine as functional implants. Finally, the best impregnation conditions based on the antioxidant capacity and phenolic composition were 6 h of impregnation, 300 bar, 35 °C, fast impregnation, 60 mg/mL concentration of extract and MSU-H silica.

## Figures and Tables

**Figure 1 materials-15-05934-f001:**
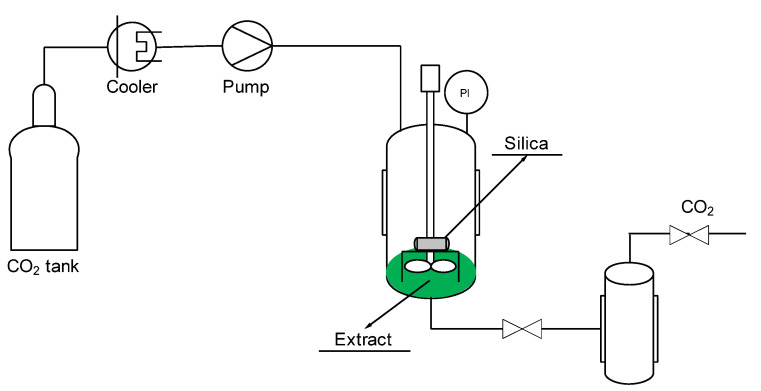
Schematic diagram of a SSI pilot plant.

**Figure 2 materials-15-05934-f002:**
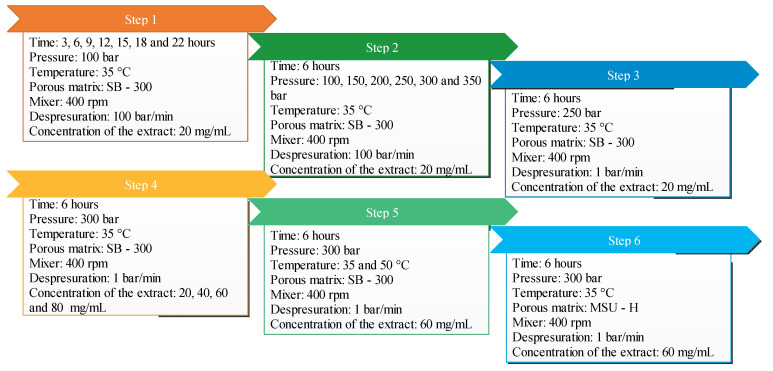
Different variables and experiments used in the impregnation process.

**Figure 3 materials-15-05934-f003:**
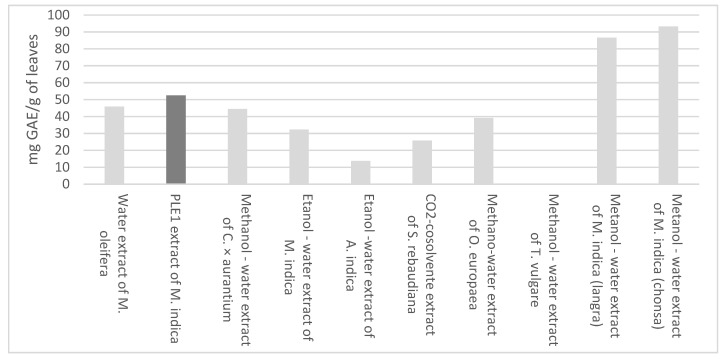
Total phenols of different extracts of leaves.

**Figure 4 materials-15-05934-f004:**
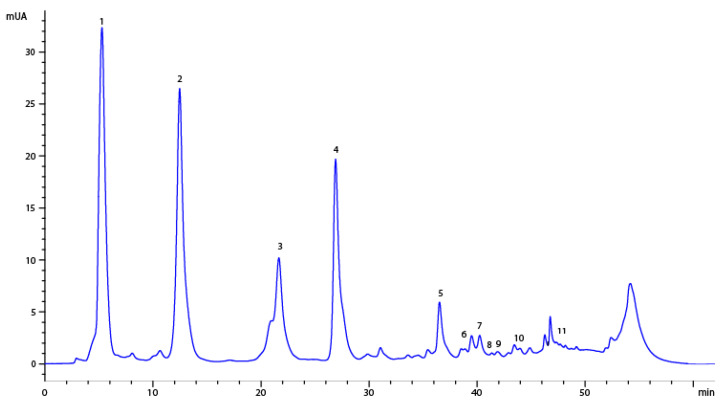
HPLC chromatogram of the extract of mango leaves. Peaks: 1, gallic acid; 2, iriflophenone 3-C-β-D-glucoside; 3, iriflophenone 3-C-(2- O-p-hydroxybenzoyl)- β -D-glucoside; 4, mangiferin; 5, iriflophenone-3-C-(2-O-galloryl)-b-D-glucoside; 6, quercetin 3-D-galactoside; 7, quercetin 3- β -D-glucoside; 8, quercetin-3-O-xyloside; 9, quercetin-3-O-β-L arabinopyranoside; 10, 1,2,3,4,6-penta-O-galloryl-β-D-glucose; 11, quercetin.

**Figure 5 materials-15-05934-f005:**
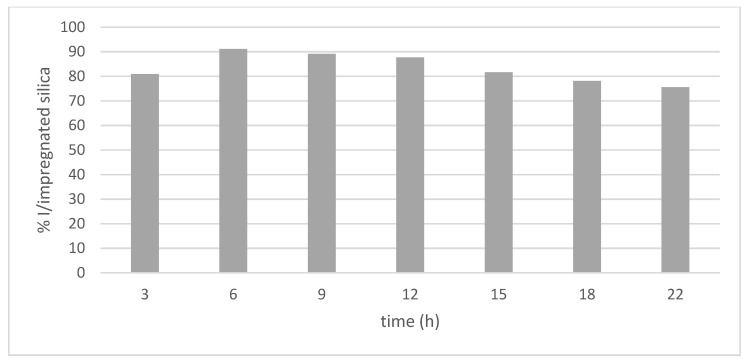
Percentage of impregnation of phenolic compounds present in mango leaves at different times.

**Figure 6 materials-15-05934-f006:**
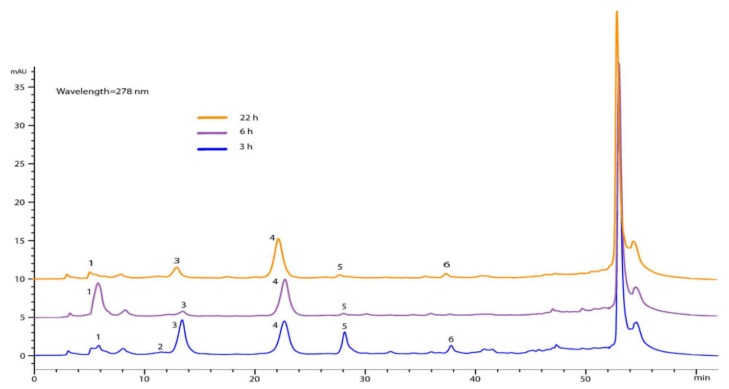
Chromatograms at different times of impregnation. (1) Gallic acid, (2) methyl gallate, (3) iriflophenone 3-C-β-D-glucoside, (4) iriflophenone 3-C-(2-o-p-hydroxybenzol)-β-D-glucoside, (5) mangiferin and (6) iriflophenone 3-C-(2-o-galloryl)-β-D-glucoside. It is worth mentioning that the process for extracting compounds from the different silicas was similar.

**Figure 7 materials-15-05934-f007:**
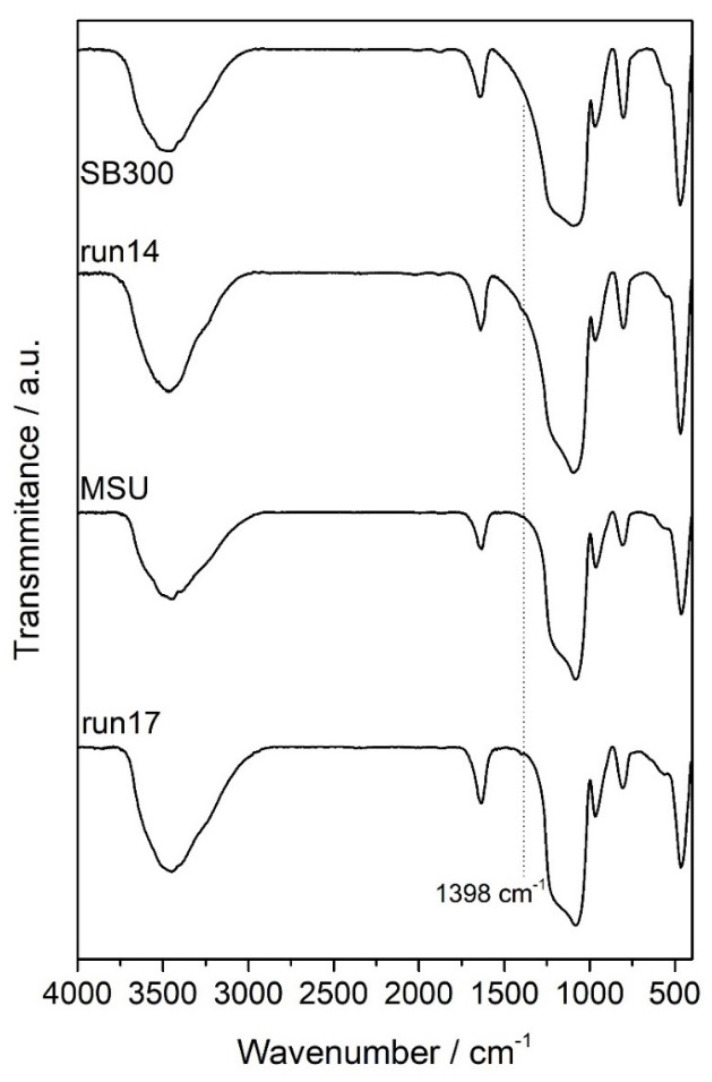
FTIR spectra of impregnated silicas.

**Figure 8 materials-15-05934-f008:**
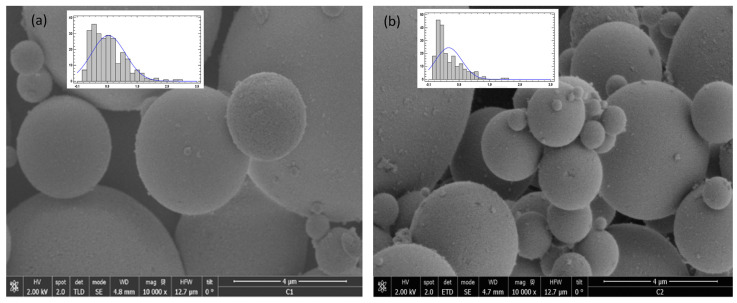
SEM image of (**a**) silica and (**b**) impregnated silica.

**Figure 9 materials-15-05934-f009:**
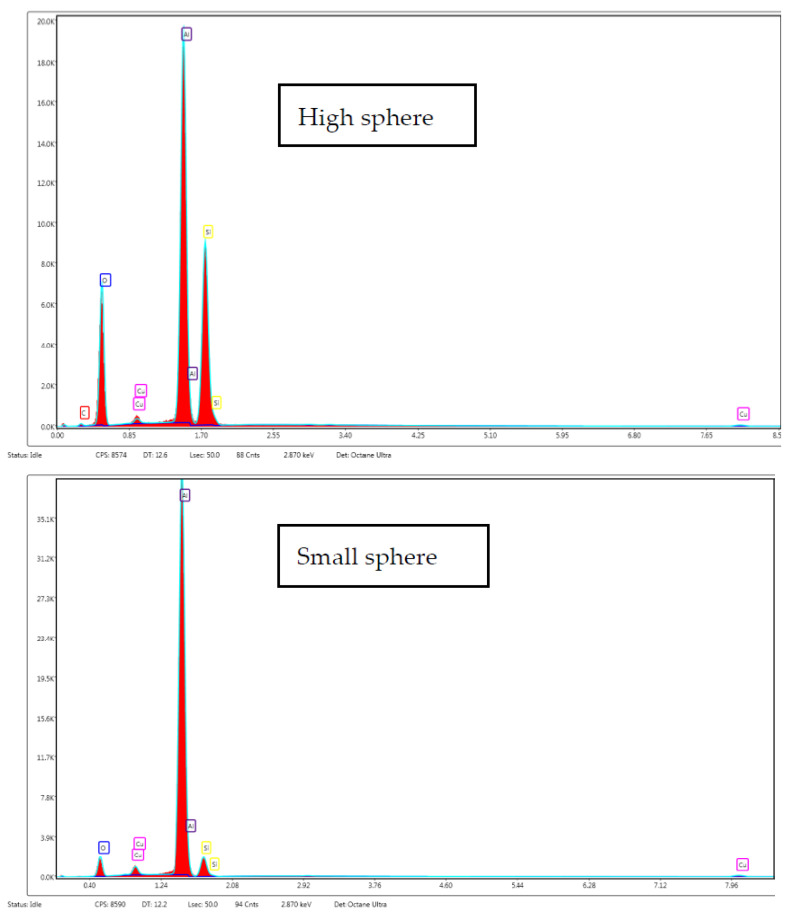
EDX spectra collected from high and small spheres.

**Figure 10 materials-15-05934-f010:**
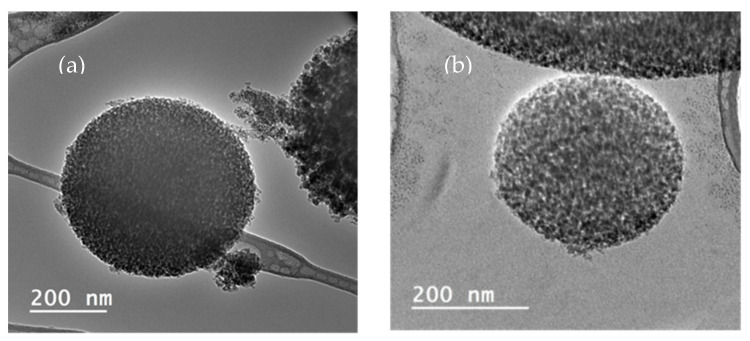
TEM image of (**a**) silica and (**b**) impregnated silica.

**Table 1 materials-15-05934-t001:** Supercritical impregnation experiments.

Experiment	Time of Impregnation (h)	Pressure (bar)	Temperature (°C)	Concentration of Extract (mg/mL)	Porous Matrix
**1**	3	100	35	20	Silica SB-300
**2**	6	100	35	20	Silica SB-300
**3**	9	100	35	20	Silica SB-300
**4**	12	100	35	20	Silica SB-300
**5**	15	100	35	20	Silica SB-300
**6**	18	100	35	20	Silica SB-300
**7**	22	100	35	20	Silica SB-300
**8**	6	150	35	20	Silica SB-300
**9**	6	200	35	20	Silica SB-300
**10**	6	250	35	20	Silica SB-300
**11 ***	6	250	35	20	Silica SB-300
**11**	6	300	35	20	Silica SB-300
**12**	6	350	35	20	Silica SB-300
**13**	6	300	35	40	Silica SB-300
**14**	6	300	35	60	Silica SB-300
**15**	6	300	35	80	Silica SB-300
**16**	6	300	50	60	MSU-H
**17**	6	300	35	60	MSU-H

* Slow depressurization (1 bar/min).

**Table 2 materials-15-05934-t002:** Number of total phenols and antioxidant capacity of mango leaves extract.

Yield %	Total Phenol mg GAE/g Extract	AAI µg DPPH/µg Extract	FRAP µmol TE/g Extract	DPPH µmol TE/g Extract
18.75	279.53 ± 3.27	2.58 ± 0.21	1898.78 ± 42.33	2637.46 ± 00.00

Yield: (g extract/g leaves) × 100; TE: Trolox equivalents; GAE: gallic acid equivalent. Values (mean ± SD) are average of one extract, analyzed in duplicate.

**Table 3 materials-15-05934-t003:** Phenolic compounds of mango leaves extract PLE1.

Gallic Acid	Iriflophenone 3-C-β-D-Glucoside	Iriflophenone 3-C-(2-O-p-Hydroxybenzoyl)-β-D-Glucoside	Mangiferin
5.84 ± 0.02	12.08 ± 0.01	5.53 ± 0.06	7.51 ± 0.09
Iriflophenone-3-C-(2-O-galloryl)-β-D-glucoside	Quercetin 3-D-galactoside	Quercetin 3-β-D-glucoside	Quercetin-3-O-xyloside
0.45 ± 0.02	0.69 ± 0.01	NQ	0.27 ± 0.02
Quercetin-3-O-a-L arabinopyranoside	1,2,3,4,6-penta-O-galloryl-β-D-glucose	Quercetin (Aglycone)	
0.18 ± 0.07	0.14 ± 0.02	0.06 ± 0.03	
mg Phenolic compounds/g of extract			

Values (mean ± SD) are the average of one extract, analyzed in duplicate.

**Table 4 materials-15-05934-t004:** Phenolic compounds at different impregnation pressures.

N°	Matrix	P (Bar)	Ce (mg/mL)	T (°C)	Phenolic Compounds mg/L	
					Gallic Acid	Iriflophenone 3-C-(2-O-P-Hydroxybenzoyl)-β-D-Glucoside
1	SB-300	100	20	35	0.83 ± 0.01 ^ab^	1.68 ± 0.03 ^a^
2		150	20	35	NQ	4.68 ± 0.53 ^bd^
3		200	20	35	0.74 ± 0.01 ^ab^	8.25 ± 0.10 ^c^
5		250 *	20	35	0.18 ± 0.04 ^c^	6.22 ± 0.10 ^bd^
6		300	20	35	1.55 ± 0.33 ^d^	15.87 ± 1.21 ^e^

* Slow depressurization (1 bar/min); NQ: not identified; P: pressure, C_e_: concentration of the extract; T: temperature; values followed by a different superscript in each column are significantly different (*p* < 0.05); values (mean ± SD) are the average of two experiments, analyzed in duplicate.

**Table 5 materials-15-05934-t005:** Antioxidant capacity and impregnated phenolic compounds from runs at different extract concentrations and temperatures.

N°	Ce (mg/mL)	T (°C)	Silica	AAI µg (DPPH/µg Antioxidant)	IC_50_ (µg Antioxidant/mL)	DPPH (µmol TE/g Silica)	FRAP (µmol TE/g Silica)	TP (mg GAE/g of Silica)	Gallic Acid (mg/L)	Iriflophenone 3-C-(2-O-P-HyDroxybenzoyl)-β-D-Glucoside (mg/L)
1	20	35	SB-300	0.30 ± 0.01 ^ae^	77.21 ± 1.36 ^a^	ND	ND	ND	1.55 ± 0.33 ^a^	15.87 ± 1.21 ^a^
2	40	35	SB-300	0.60 ± 0.03 ^b^	35.53 ± 1.66 ^b^	ND	ND	ND	0.45 ± 0.13 ^b^	15.79 ± 3.18 ^a^
3	60	35	SB-300	0.74 ± 0.01 ^cd^	28.64 ± 0.39 ^c^	ND	ND	ND	2.52 ± 0.18 ^c^	35.62 ± 0.19 ^b^
4	80	35	SB-300	0.44 ± 0.05 ^be^	54.16 ± 5.97 ^c^	4.63 ± 0.11	43.04 ± 0.07	2.51 ± 0.00	4.54 ± 0.02 ^d^	44.06 ± 0.28 ^c^
5	60	50	SB-300	0.37 ± 0.00 ^ae^	65.67 ± 0.42 ^ac^	ND	ND	ND	2.73 ± 0.14 ^c^	30.85 ± 0.39 ^d^
6	60	35	MSU-H	1.05 ± 0.13 ^d^	17.28 ± 1.64 ^b^	1.80 ± 0.08	62.27 ± 1.75	4.32 ± 0.95	7.08 ± 0.25 ^e^	83.33 ± 0.46 ^e^
7	60	50	MSU-H	0.15 ± 0.02 ^e^	140.11 ± 15.28 ^d^	ND	ND	ND	-	-

Pressure: 300 bar; TP: total phenols; ND: the analysis of DPPH, FRAP and TP was only performed on run 4 and 6. Values followed by a different superscript in each column are significantly different (*p* < 0.05); values (mean ± SD) are the average of two experiments, analyzed in duplicate.

## Data Availability

Not applicable.

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
