# Peer review of "Inclusion of Natural Antioxidants of Mango Leaves in Porous Ceramic Matrices by Supercritical CO_2_ Impregnation"

_materials, 2022, doi:10.3390/ma15175934_

Round 1
Reviewer 1 Report
1. Introduction
L 32-36 incorrect wording: against diphtheria and rheumatism, the cooked fruit against diarrhea and chronic dysentery. “In the treatment of …” would be the correct wording.
2. Materials and Methods
L 113 to be corrected: Mangifera indicates L. leaves (Kent variety) extract was collected in 2019…..
In the case of the DPPH method, describe the specifications for determining the AA in the native extract. Describe the composition of the sample in the case of the ERAP method and the Total Phenolic content.
3. Results and Discussion
L 283-293 It’s useless to compare the AA of mango leaves extract with AA of extracts obtained from other plants, as long as it’s not specified on which surfaces or porous structures have been impregnated with the research topic.
Here’s my proposal:
1) The analysis under this aspect or
2) The comparison of AA extracts from other parts of the mango plant
3) The removal of figure 3 from the text.
Figure 4 and description. I propose maintaining the research theme. That’s why it should be analyzed only the component of phenols in the extracts obtained from mango (they are sufficient) .
Table 5. Strangely, there are no values for AA by the DPPH method when there are values for AAI. Also, for phenols: gallic acid and Iriflophenone 3- C-(2-O-phydroxybenzoyl)-β-D-glucoside were quantitatively determined and the Total Phenolic content test shows the absence of phenols.
Here’s my proposal:
A) To eliminate the indicators of antioxidant activity from table 5. For discussions to make references to another table with the antioxidant activity of extracts impregnated in optimal thermal conditions or
B) A recalculation for DPPH based on the absorbance that was already determined, or
2) Modifying the parameters for the ERAP method and Total Phenolic content to increase sensitivity, concentrate the samples or 3) conduct an elementary mathematical recalculation.
Author Response
- Introduction
L 32-36 incorrect wording: against diphtheria and rheumatism, the cooked fruit against diarrhea and chronic dysentery. “In the treatment of …” would be the correct wording.
As the reviewer suggests the wording has been corrected
- Materials and Methods
L 113 to be corrected: Mangifera indicates L. leaves (Kent variety) extract was collected in 2019…..
As the reviewer suggests this sentence has been corrected
In the case of the DPPH method, describe the specifications for determining the AA in the native extract. Describe the composition of the sample in the case of the ERAP method and the Total Phenolic content.
As the reviewer suggests this fact has been better described in the manuscript in the sections 2.4.1, 2.4.2 and 2.5.1
- Results and Discussion
L 283-293 It’s useless to compare the AA of mango leaves extract with AA of extracts obtained from other plants, as long as it’s not specified on which surfaces or porous structures have been impregnated with the research topic. Here’s my proposal:
1) The analysis under this aspect or
2) The comparison of AA extracts from other parts of the mango plant
3) The removal of figure 3 from the text.
As the reviewer suggests authors have removed Figure 3 from the text
Figure 4 and description. I propose maintaining the research theme. That’s why it should be analyzed only the component of phenols in the extracts obtained from mango (they are sufficient)
As the reviewer suggest Figure 4 has been maintained
Table 5. Strangely, there are no values for AA by the DPPH method when there are values for AAI. Also, for phenols: gallic acid and Iriflophenone 3- C-(2-O-phydroxybenzoyl)-β-D-glucoside were quantitatively determined and the Total Phenolic content test shows the absence of phenols.Here’s my proposal:
- A) To eliminate the indicators of antioxidant activity from table 5. For discussions to make references to another table with the antioxidant activity of extracts impregnated in optimal thermal conditions or
- B) A recalculation for DPPH based on the absorbance that was already determined, or 2) Modifying the parameters for the ERAP method and Total Phenolic content to increase sensitivity, concentrate the samples or 3) conduct an elementary mathematical recalculation.
DPPH, FRAP and TP were carried out to runs 4 and 6. For that reason there are no data in the rest of columns. These experiments were selected due to they had the same operating conditions and different silica. Footnotes of table 5 has been included to clarify the process
Reviewer 2 Report
I think the manuscript entitled “Inclusion of natural antioxidants of mango leaves in porous ceramic matrices by supercritical CO2 impregnation” is interesting. However, authors need to present the data in a better format. Therefore, I think the manuscript must go under major revision before acceptance.
1. The scientific name of the plant must be written as per the rule of scientific nomenclature. I suggest modification as per the guidelines in the manuscript (L113).
2. I wonder if authors have performed any statistical analysis of the data?
3. How many times has the experiment been performed? Error bar is missing from all the figures!
4. Authors must write the section for the statistics in the methodology section of the manuscript.
5. Author must improve the methodology and discussion part of the manuscript.
6. Authors must identify active compounds present in the extract and provide the HPLC data in the main figure.
7. I recommend writing a paragraph more about the supercritical CO2 as an alternative and effective strategy. The below mentioned papers are suitable for citation:
Kavya et al., 2022. Colloids and Surfaces B: Biointerfaces.216: 112584.
Sanchez-Sanchez et al., 2017. The Journal of Supercritical Fluids. 128: 208-217.
Villacís-Chiriboga et al., 2021. Foods. 10: 2201.
Author Response
Dear reviewers:
Thank you very much for your consideration, and we really appreciate the comments and have learned a lot. Appropriate changes were made and highlighted in the revised manuscript according to the suggestions of reviewers and editor.
Reviewer #2:
I think the manuscript entitled “Inclusion of natural antioxidants of mango leaves in porous ceramic matrices by supercritical CO2 impregnation” is interesting. However, authors need to present the data in a better format. Therefore, I think the manuscript must go under major revision before acceptance.
- The scientific name of the plant must be written as per the rule of scientific nomenclature. I suggest modification as per the guidelines in the manuscript (L113).
As the reviewer suggests scientific name format of the plant has been updated
- I wonder if authors have performed any statistical analysis of the data?
An ANOVA of AAI data and phenolic compounds in the experiments using different concentration, temperature and pressure were included. However, when authors analysed different times of impregnation, a statistical analysis is missed due to did one repetition (Figure 5). These analyses were done at the beginning of experimentation and were carried out as screening to define the impregnation time and for that reason were not done several times each.
- How many times has the experiment been performed? Error bar is missing from all the figures!
Experiments of Figure 4 were carried out twice and the analyses the same. An error bar has been included in Figure 4. Anyway in Figure 5 error bar were not included due to the experiments were carried out only one time
- Authors must write the section for the statistics in the methodology section of the manuscript.
As the reviewer suggests the statistics has been included and written
- Author must improve the methodology and discussion part of the manuscript.
Authors have tried to improve these parts
- Authors must identify active compounds present in the extract and provide the HPLC data in the main figure.
As the reviewer suggests a chromatogram of the extract has been included
7.I recommend writing a paragraph more about the supercritical CO2 as an alternative and effective strategy. The below mentioned papers are suitable for citation:
Kavya et al., 2022. Colloids and Surfaces B: Biointerfaces.216: 112584.
Sanchez-Sanchez et al., 2017. The Journal of Supercritical Fluids. 128: 208-217.
Villacís-Chiriboga et al., 2021. Foods. 10: 2201.
As the reviewer suggests a paragraph has been included and some of these references were cited
Reviewer 3 Report
The paper is precise and well written. I recommend it for publication in the present form.
Author Response
Reviewer suggest the paper is precise and well written and recommend it for publication in the present form. Anyway the paper has been improved for it
Round 2
Reviewer 2 Report
The revision is not satisfactory. The manuscript need more scientific attention.
I wonder how many times authors have repeated the experiment?
Why there is no error bars in figure 3 and figure5? How many times the experience was performed?
The data is very good to be true in several cases. For example, I doubt, if it is possible to get zero standard deviation even in technical replicates. I don’t know how authors got “2637.46 ± 00.00” Value in DPPH μmol TE/g extract (in Table 2).
Authors have not followed the rules for scientific nomenclature, these shows the scientific negligence.
The quality of HPLC data depicted in figure 4 is very poor. Axis is not visible.
I recommend to indicate the number of replicates used to conclude the data in each figure description.
Author Response
Comments and Suggestions for Authors
The revision is not satisfactory. The manuscript need more scientific attention.
I wonder how many times authors have repeated the experiment?
The experiments of impregnation with different pressures, concentrations and type of silica were carried out twice as total phenolic content and antioxidant activity
Why there is no error bars in figure 3 and figure5? How many times the experience was performed?
In Figure 3, phenolic total result of PLE extract belong only to this investigation. Other data belong to other investigations. In the figure does not show the error bar of total phenolic of extract but in the manuscript in lines 312 is shown. The figure was changed and in the manuscript was added in line 312 the value in mg GAE/g leaves to compare with other investigations
The experiments of Figure 5 were not done more than one time. Replicas were carried out in the experiments where pressure, concentration and type silica were evaluated
The data is very good to be true in several cases. For example, I doubt, if it is possible to get zero standard deviation even in technical replicates. I don’t know how authors got “2637.46 ± 00.00” Value in DPPH μmol TE/g extract (in Table 2).
Data are right. In this case, two measures of antioxidant activity of extract PLE1 were carried out by DPPH method. As the two absorbance of the same extract were identical, the final valour showed in units of (µmol TE/g extract) are the same. If reviewer considers that is needed, we could send Excel file where all the calculation was made.
|
Sample |
Solvent |
P bar |
Ppm real |
Abs (nm) |
Co preliminar (mg/L) |
Ce BS(umol TE/g extract) |
|
|
PLE 1 |
EtOH |
120 |
87,5 |
0,706 |
230,78 |
2637,46 |
2637,46 ± 0,00 |
|
0,706 |
230,78 |
2637,46 |
|||||
Authors have not followed the rules for scientific nomenclature, these shows the scientific negligence.
Authors do not understand very well this comment. We observed some mistake as “hours” instead of “h” that it has been corrected. Anyway if reviewer considers other mistakes please indicate where are…
The quality of HPLC data depicted in figure 4 is very poor. Axis is not visible.
As the reviewers suggest the quality of image was improved
I recommend to indicate the number of replicates used to conclude the data in each figure description.
As the reviewer suggests a footnote, in each table and figure that was possible, were included.
“Values (mean ± SD) are average of one extract, analyzed in duplicate”
Anyway in Figure 3 was not included due the data belong to other researcher and in Table 2 it is already commented that two replicas of total phenolic content and antioxidant activity were done. In Figure 4 does not appear due to in Table 3 it is already described it. In Figure 5 was not included due to repetitions were not made. The same in Figures 7-10.
